# Ancient Egyptian mummy genomes suggest an increase of Sub-Saharan African ancestry in post-Roman periods

Verena J. Schuenemann[1,2,*], Alexander Peltzer[3,4,*], Beatrix Welte[1], W. Paul van Pelt[5], Martyna Molak[6], Chuan-Chao Wang[4], Anja Furtwängler[1], Christian Urban[1], Ella Reiter[1], Kay Nieselt[3], Barbara Teßmann[7], Michael Francken[1], Katerina Harvati[1,2,8], Wolfgang Haak[4,9], Stephan Schiffels[4] & Johannes Krause[1,2,4]

Egypt, located on the isthmus of Africa, is an ideal region to study historical population dynamics due to its geographic location and documented interactions with ancient civilizations in Africa, Asia and Europe. Particularly, in the first millennium BCE Egypt endured foreign domination leading to growing numbers of foreigners living within its borders possibly contributing genetically to the local population. Here we present 90 mitochondrial genomes as well as genome-wide data sets from three individuals obtained from Egyptian mummies. The samples recovered from Middle Egypt span around 1,300 years of ancient Egyptian history from the New Kingdom to the Roman Period. Our analyses reveal that ancient Egyptians shared more ancestry with Near Easterners than present-day Egyptians, who received additional sub-Saharan admixture in more recent times. This analysis establishes ancient Egyptian mummies as a genetic source to study ancient human history and offers the perspective of deciphering Egypt's past at a genome-wide level.

[1] Institute for Archaeological Sciences, University of Tübingen, 72070 Tübingen, Germany. [2] Senckenberg Centre for Human Evolution and Palaeoenvironment, University of Tübingen, 72070 Tübingen, Germany. [3] Integrative Transcriptomics, Center for Bioinformatics, University of Tübingen, 72076 Tübingen, Germany. [4] Department for Archaeogenetics, Max Planck Institute for the Science of Human History, 07745 Jena, Germany. [5] Division of Archaeology, University of Cambridge, Cambridge CB2 3DZ, UK. [6] Museum and Institute of Zoology, Polish Academy of Sciences, 00-679 Warsaw, Poland. [7] Berlin Society of Anthropology, Ethnology and Prehistory, 10997 Berlin, Germany. [8] DFG Centre for Advanced Studies 'Words, Bones, Genes, Tools: Tracking Linguistic, Cultural and Biological Trajectories of the Human Past', University of Tübingen, 72070 Tübingen, Germany. [9] School of Biological Sciences, The University of Adelaide, Adelaide, South Australia 5005, Australia. * These authors contributed equally to this work. Correspondence and requests for materials should be addressed to W.H. (email: haak@shh.mpg.de) or to S.S. (email: schiffels@shh.mpg.de) or to J.K. (email: krause@shh.mpg.de).

Egypt provides a privileged setting for the study of population genetics as a result of its long and involved population history. Owing to its rich natural resources and strategic location on the crossroads of continents, the country had intense, historically documented interactions with important cultural areas in Africa, Asia and Europe ranging from international trade to foreign invasion and rule. Especially from the first millennium BCE onwards, Egypt saw a growing number of foreigners living and working within its borders and was subjected to an almost continuous sequence of foreign domination by Libyans, Assyrians, Kushites, Persians, Greeks, Romans, Arabs, Turks and Brits. The movement of people, goods and ideas throughout Egypt's long history has given rise to an intricate cultural and genetic exchange and entanglement, involving themes that resonate strongly with contemporary discourse on integration and globalization[1].

Until now the study of Egypt's population history has been largely based on literary and archaeological sources and inferences drawn from genetic diversity in present-day Egyptians. Both approaches have made crucial contributions to the debate but are not without limitations. On the one hand, the interpretation of literary and archaeological sources is often complicated by selective representation and preservation and the fact that markers of foreign identity, such as, for example, Greek or Latin names and ethnics, quickly became 'status symbols' and were adopted by natives and foreigners alike[2–4]. On the other hand, results obtained by modern genetic studies are based on extrapolations from their modern data sets and make critical assumptions on population structure and time[5]. The analysis of ancient DNA provides a crucial piece in the puzzle of Egypt's population history and can serve as an important corrective or supplement to inferences drawn from literary, archaeological and modern DNA data.

Despite their potential to address research questions relating to population migrations, genetic studies of ancient Egyptian mummies and skeletal material remain rare, although research on Egyptian mummies helped to pioneer the field of ancient DNA research with the first reported retrieval of ancient human DNA[6]. Since then progress has been challenged by issues surrounding the authentication of the retrieved DNA and potential contaminations inherent to the direct PCR method[7]. Furthermore, the potential DNA preservation in Egyptian mummies was met with general scepticism: The hot Egyptian climate, the high humidity levels in many tombs and some of the chemicals used in mummification techniques, in particular sodium carbonate, all contribute to DNA degradation and are thought to render the long-term survival of DNA in Egyptian mummies improbable[8]. Experimental DNA decay rates in papyri have also been used to question the validity and general reliability of reported ancient Egyptian DNA results[9]. The recent genetic analysis of King Tutankhamun's family[10] is one of the latest controversial studies that gave rise to this extensive scholarly debate[11]. New data obtained with high-throughput sequencing methods have the potential to overcome the methodological and contamination issues surrounding the PCR method and could help settle the debate surrounding ancient Egyptian DNA preservation[8]. However, the first high-throughput sequences obtained from ancient Egyptian mummies[12] were not supported by rigorous authenticity and contamination tests.

Here, we provide the first reliable data set obtained from ancient Egyptians using high-throughput DNA sequencing methods and assessing the authenticity of the retrieved ancient DNA via characteristic nucleotide misincorporation patterns[13,14] and statistical contamination tests[15] to ensure the ancient origin of our obtained data.

By directly studying ancient DNA from ancient Egyptians, we can test previous hypotheses drawn from analysing modern Egyptian DNA, such as recent admixture from populations with sub-Saharan[16] and non-African ancestries[17], attributed to trans-Saharan slave trade and the Islamic expansion, respectively. On a more local scale, we aim to study changes and continuities in the genetic makeup of the ancient inhabitants of the Abusir el-Meleq community (Fig. 1), since all sampled remains derive from this community in Middle Egypt and have been radiocarbon dated to the late New Kingdom to the Roman Period (cal. 1388BCE–426CE, Supplementary Data 1). In particular, we seek to determine if the inhabitants of this settlement were affected at the genetic level by foreign conquest and domination, especially during the Ptolemaic (332–30BCE) and Roman (30BCE–395CE) Periods.

## Results

**Samples and anthropological analysis.** All 166 samples from 151 mummified individuals (for details of the 90 individuals included in the later analysis, see Supplementary Data 1) used in this study were taken from two anthropological collections at the University of Tübingen and the Felix von Luschan Skull Collection, which is now kept at the Museum of Prehistory of the Staatliche Museen zu Berlin, Stiftung preußischer Kulturbesitz (individuals: S3533, S3536, S3544, S3552, S3578, S3610). According to the radiocarbon dates (Supplementary Data 1, see also ref. 18), the samples can be grouped into three time periods: Pre-Ptolemaic (New Kingdom, Third Intermediate Period and Late Period), Ptolemaic and Roman Period. During their conservation in the Tübingen and Berlin collections the remains underwent different treatments: some were preserved in their original mummified state, while others were macerated for anthropological analysis or due to conservation problems[19].

In most cases, non-macerated mummy heads still have much of their soft tissue preserved. Some of the remains (individuals analysed in our study: 1543, 1547, 1565, 1577, 1611) have traces of gold leaf near the mouth and the cheekbone, which is characteristic for mummies from the Ptolemaic Period onwards[20]. In most cases the brain was removed and the excerebration route was highly likely transnasal, resulting in visible defects on the cribriform plate (for the individuals analysed in our study, see Supplementary Data 1). In summary, the excellent bone preservation and the more or less good soft tissue preservation made a wide-ranging analysis possible[19].

Recently, various studies were conducted on these remains, including a study on ancient Egyptian embalming resins, two ancient DNA studies and an anthropological examination of the macerated crania[12,18,19,21]. While the possibilities of a demographic reconstruction based on anthropological finds are naturally limited—due to incompleteness of the assemblage, the following anthropological observations were made on the assemblage: For a first assessment, computer tomographic scans of 30 mummies with soft tissue preservation were produced to describe sex (Supplementary Data 1), age at death (Supplementary Data 1) and the macroscopic health status; the six macerated mummies were examined directly. It is notable that most of the individuals are early and late adults, and that subadult individuals are underrepresented (Supplementary Data 1). It is possible that the sample's demographic profile is the result of different burial treatments for adults and subadults, but it seems more likely that it is due to collection bias, with collectors favouring intact adult skulls. Almost all of the teeth show significant dentine exposure up to a total loss of the crown. This abrasion pattern is likely due to the food and food preparation itself, in particular for a cereal-rich diet containing a high proportion of coarse sandy particles. These particles act to abrade the dental tissues, allowing bacteria to penetrate the interior of the teeth. As a result, carious lesions or periapical processes appear in the analysed individuals (Supplementary Data 1)[19].

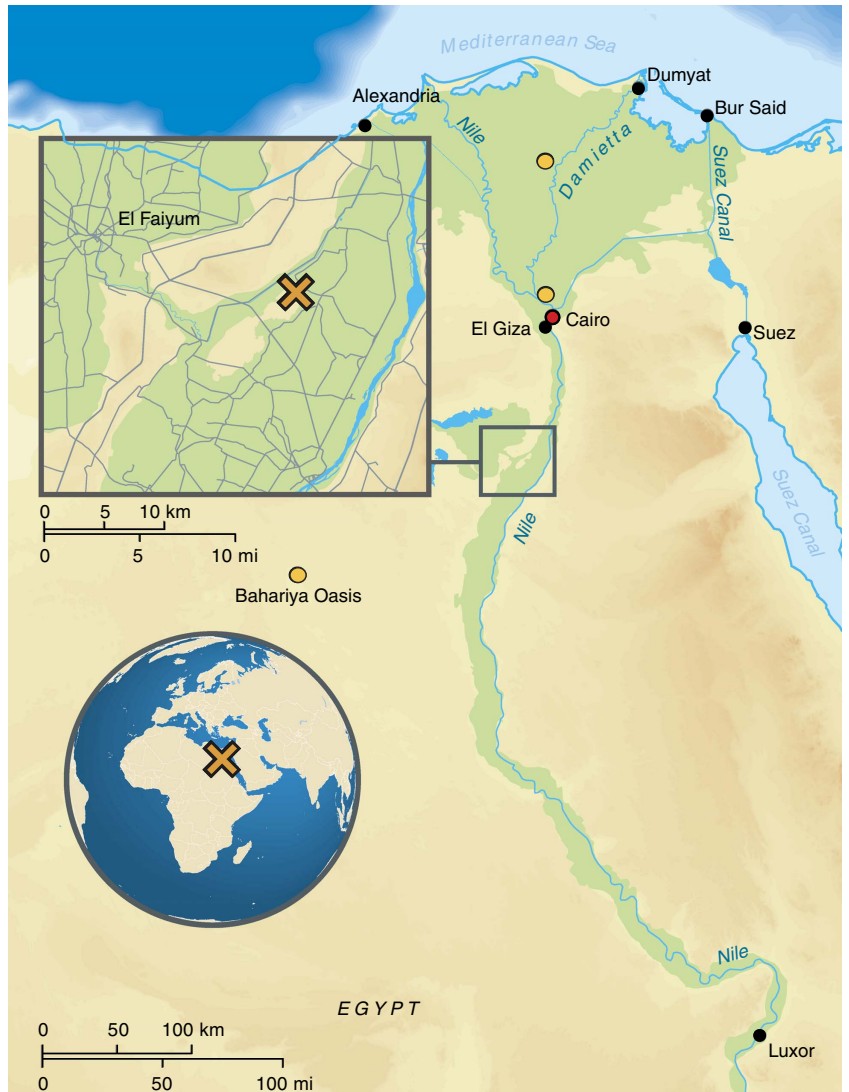

**Figure 1 | Geographic context, of the samples used in this study.** Map of Egypt depicting the location of the archaeological site Abusir-el Meleq (orange X) and the location of the modern Egyptian samples (orange circles) (design of the graphic by Annette Günzel).

For the DNA analysis we sampled different tissues (bone, soft tissue, tooth), macerated and non-macerated, to test for human DNA preservation.

**Processing and sequencing of the samples**. We extracted DNA from 151 mummified human remains and prepared double-stranded Illumina libraries with dual barcodes[22,23]. Then we used DNA capture techniques for human mitochondrial DNA[24] and for 1.24 million genomic single nucleotide polymorphisms (SNPs)[25] in combination with Illumina sequencing, through which we successfully obtained complete human mitochondrial genomes from 90 samples and genome-wide SNP data from three male individuals passing quality control.

**Comparison of the DNA preservation in different tissues**. We tested different tissues for DNA preservation and applied strict criteria for authenticity on the retrieved mitochondrial and nuclear DNA to establish authentic ancient Egyptian DNA. First, DNA extracts from several tissues (that is, bone, teeth, soft tissue and macerated teeth) from 151 individuals were screened for the presence of human mitochondrial DNA (mtDNA) resulting in a total of 2,157 to 982,165 quality filtered mitochondrial reads per

sample, and 11- to 4,236-fold coverage. To estimate, identify and filter out potential contamination we applied the program schmutzi[15] with strict criteria for contamination and kept only samples with less than 3% contamination for further analysis. For a comparison of different source material (soft tissue, bone and teeth) ten individuals (Supplementary Table 1) were sampled multiple times. Yields of preserved DNA were comparable in bone and teeth but up to ten times lower in soft tissues (Fig. 2a, Supplementary Table 1). Nucleotide misincorporation patterns characteristic for damaged ancient human DNA allowed us to assess the authenticity of the retrieved DNA[13,14]. The observed DNA damage patterns differed for the source materials with on average 19% damage in soft tissues and around 10% damage in bone tissue and teeth (Fig. 2b,c, Supplementary Table 1). Importantly, mtDNA haplotypes were identical for all samples from the same individuals. Our results thus suggest that DNA damage in Egyptian mummies correlates with tissue type. The protection of bone and teeth by the surrounding soft tissue or the embalmment of soft tissue may have contributed to the observed differences.

**Generation of nuclear data**. In order to analyse the nuclear DNA we selected 40 samples with high mtDNA coverage and low

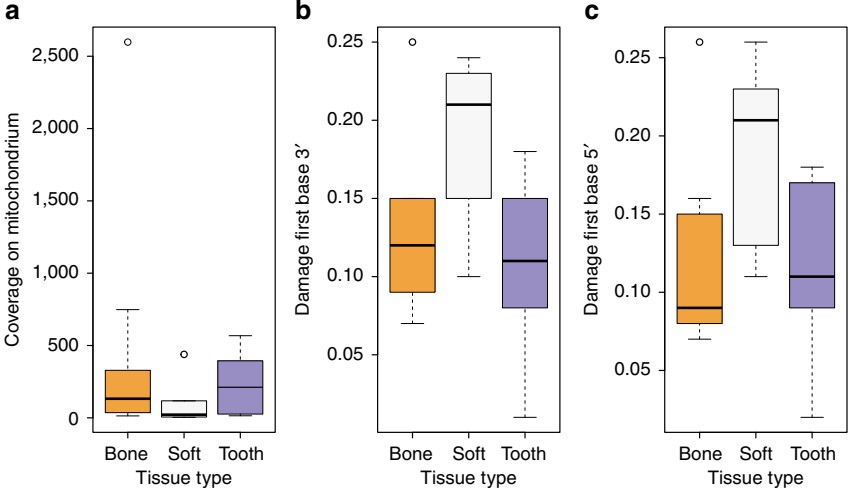

**Figure 2 | DNA preservation and DNA damage of the samples used in this study.** (**a**) coverage boxplots separated by tissue type (bone, mummified tissue, teeth), (**b**) boxplots showing damage of first base at the 3′ end separated by tissue type according to **a**, (**c**) damage on first base at the 5′ end of mapped reads separated by tissue type according to **a** and **b**.

mtDNA contamination. Using in solution enrichment for 1.2 million genome-wide SNPs[26], we obtained between 3,632 and 508,360 target SNPs per sample (Supplementary Data 2). Overall, the nuclear DNA showed poor preservation compared to the mtDNA as depicted by a high mitochondrial/nuclear DNA ratio of on average around 18,000. In many samples, nuclear DNA damage was relatively low, indicating modern contamination. We sequenced two libraries per sample: one untreated library to assess DNA damage, and one library treated with enzymatic damage repair[27], which was used for downstream analysis. We applied strict criteria for further analysis: we considered only male samples with at least 8% average cytosine deamination rates at the ends of the reads from the untreated library, and with at least 150 SNPs on the X chromosome covered at least twice, in order to estimate contamination levels reliably. Three out of 40 samples fulfilling these criteria had acceptable nuclear contamination rates: Two samples from the Pre-Ptolemaic Periods (New Kingdom to Late Period) had 5.3 and 0.5% nuclear contamination and yielded 132,084 and 508,360 SNPs, respectively, and one sample from the Ptolemaic Period had 7.3% contamination and yielded 201,967 SNPs. As shown below, to rule out any impact of potential contamination on our results, we analysed the three samples separately or replicated results using only the least contaminated sample.

**Analysis of mitochondrial genomes**. The 90 mitochondrial genomes fulfilling our criteria (>10-fold coverage and <3% contamination) were grouped into three temporal categories based on their radiocarbon dates (Supplementary Data 1), corresponding to Pre-Ptolemaic Periods ($n = 44$), the Ptolemaic Period ($n = 27$) and the Roman Period ($n = 19$) (Supplementary Data 1). To test for genetic differentiation and homogeneity we compared haplogroup composition, calculated $F_{ST}$-statistics[28] and applied a test for population continuity[29] (Supplementary Table 2, Supplementary Data 3,4) on mitochondrial genome data from the three ancient and two modern-day populations from Egypt and Ethiopia, published by Pagani and colleagues[17], including 100 modern Egyptian and 125 modern Ethiopian samples (Fig. 3a). We furthermore included data from the El-Hayez oasis published by Kujanová and colleagues[30]. We observe highly similar haplogroup profiles between the three ancient groups (Fig. 3a), supported by low $F_{ST}$ values (<0.05) and P values >0.1 for the continuity test.

Modern Egyptians share this profile but in addition show a marked increase of African mtDNA lineages L0–L4 up to 20% (consistent with nuclear estimates of 80% non-African ancestry reported in Pagani et al.[17]). Genetic continuity between ancient and modern Egyptians cannot be ruled out by our formal test despite this sub-Saharan African influx, while continuity with modern Ethiopians[17], who carry >60% African L lineages, is not supported (Supplementary Data 5). To further test genetic affinities and shared ancestry with modern-day African and West Eurasian populations we performed a principal component analysis (PCA) based on haplogroup frequencies and Multidimensional Scaling of pairwise genetic distances. We find that all three ancient Egyptian groups cluster together (Fig. 3b), supporting genetic continuity across our 1,300-year transect. Both analyses reveal higher affinities with modern populations from the Near East and the Levant compared to modern Egyptians (Fig. 3b,c). The affinity to the Middle East finds further support by the Y-chromosome haplogroups of the three individuals for which genome-wide data was obtained, two of which could be assigned to the Middle-Eastern haplogroup J, and one to haplogroup E1b1b1 common in North Africa (Supplementary Table 3). However, comparative data from a contemporary population under Roman rule in Asia Minor, from the Roman city Ağlasun today in Turkey[31], did not reveal a closer relationship to the ancient Egyptians from the Roman period (Fig. 3b,c).

**Population size estimation using BEAST**. The finding of a continuous population through time allowed us to estimate the effective population size ($N_e$) from directly radiocarbon-dated mitochondrial genomes using BEAST[32]. Our results show similar values of effective population size in the different ancient time periods with an average value of between ca. 48,000 and 310,000 (average 95% CI) inhabitants in the region and period under investigation (Fig. 3d, Supplementary Fig. 2, Supplementary Table 4). This is important as it is the first time that such estimates can be contrasted with reported historic Egyptian census numbers from the neighbouring Fayum in the early Ptolemaic Period, which had a reported total population size of 85,000–95,000 inhabitants[33].

**Population genetic analysis of nuclear DNA**. On the nuclear level we merged the SNP data of our three ancient individuals

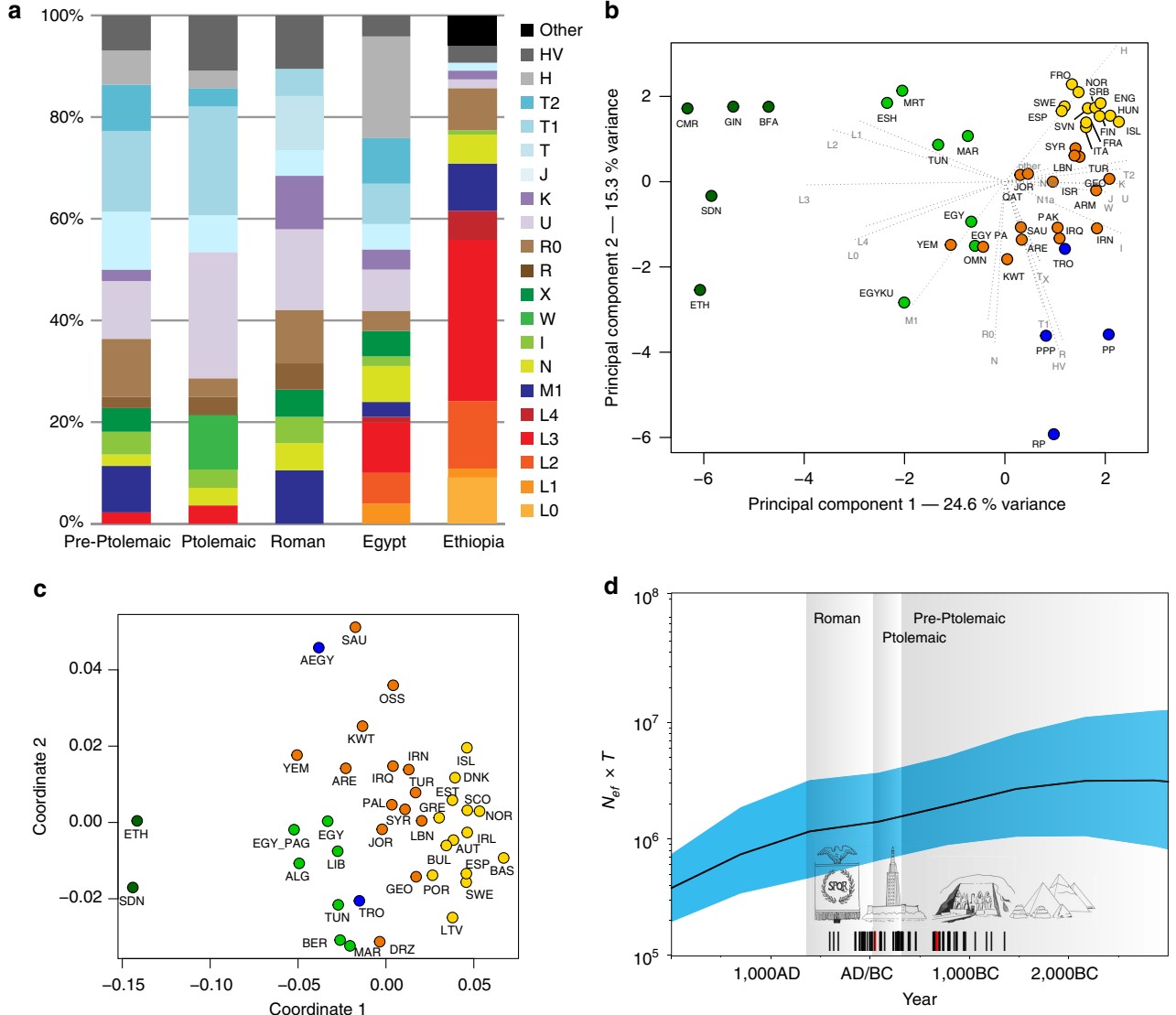

**Figure 3 | Analysis of 90 ancient Egyptian mitochondrial genomes.** (**a**) Mitochondrial DNA haplogroup frequencies of three ancient and two modern-day populations, (**b**) Principal Component Analysis based on haplogroup frequencies: (sub-Saharan Africa (green), North Africa (light green), Near East (orange), Europe (yellow), ancient (blue)), (**c**) MDS of HVR-I sequence data: colour scheme as above; note that ancient groups were pooled, (**d**) Skygrid plot depicting effective population size estimates over the last 5,000 years in Egypt. Vertical bars indicate the ages of the analysed 90 mitochondrial genomes (three samples with genome-wide data highlighted in red). Note that the values on y axis are given in female effective population size times generation time and were rescaled by 1:14.5 for the estimation of the studied population size (assuming 29-year generation time and equal male and female effective population sizes) (images by Kerttu Majander).

with 2,367 modern individuals[34,35] and 294 ancient genomes[36] and performed PCA on the joined data set. We found the ancient Egyptian samples falling distinct from modern Egyptians, and closer towards Near Eastern and European samples (Fig. 4a, Supplementary Fig. 3, Supplementary Table 5). In contrast, modern Egyptians are shifted towards sub-Saharan African populations. Model-based clustering using ADMIXTURE[37] (Fig. 4b, Supplementary Fig. 4) further supports these results and reveals that the three ancient Egyptians differ from modern Egyptians by a relatively larger Near Eastern genetic component, in particular a component found in Neolithic Levantine ancient individuals[36] (Fig. 4b). In contrast, a substantially larger sub-Saharan African component, found primarily in West-African Yoruba, is seen in modern Egyptians compared to the ancient samples. In both PCA and ADMIXTURE analyses, we did not find significant differences between the three ancient

samples, despite two of them having nuclear contamination estimates over 5%, which indicates no larger impact of modern DNA contamination. We used outgroup f3-statistics[38] (Fig. 5a,b) for the ancient and modern Egyptians to measure shared genetic drift with other ancient and modern populations, using Mbuti as outgroup. We find that ancient Egyptians are most closely related to Neolithic and Bronze Age samples in the Levant, as well as to Neolithic Anatolian and European populations (Fig. 5a,b). When comparing this pattern with modern Egyptians, we find that the ancient Egyptians are more closely related to all modern and ancient European populations that we tested (Fig. 5b), likely due to the additional African component in the modern population observed above. By computing f3-statistics[38], we determined whether modern Egyptians could be modelled as a mixture of ancient Egyptian and other populations. Our results point towards sub-Saharan African populations as the missing

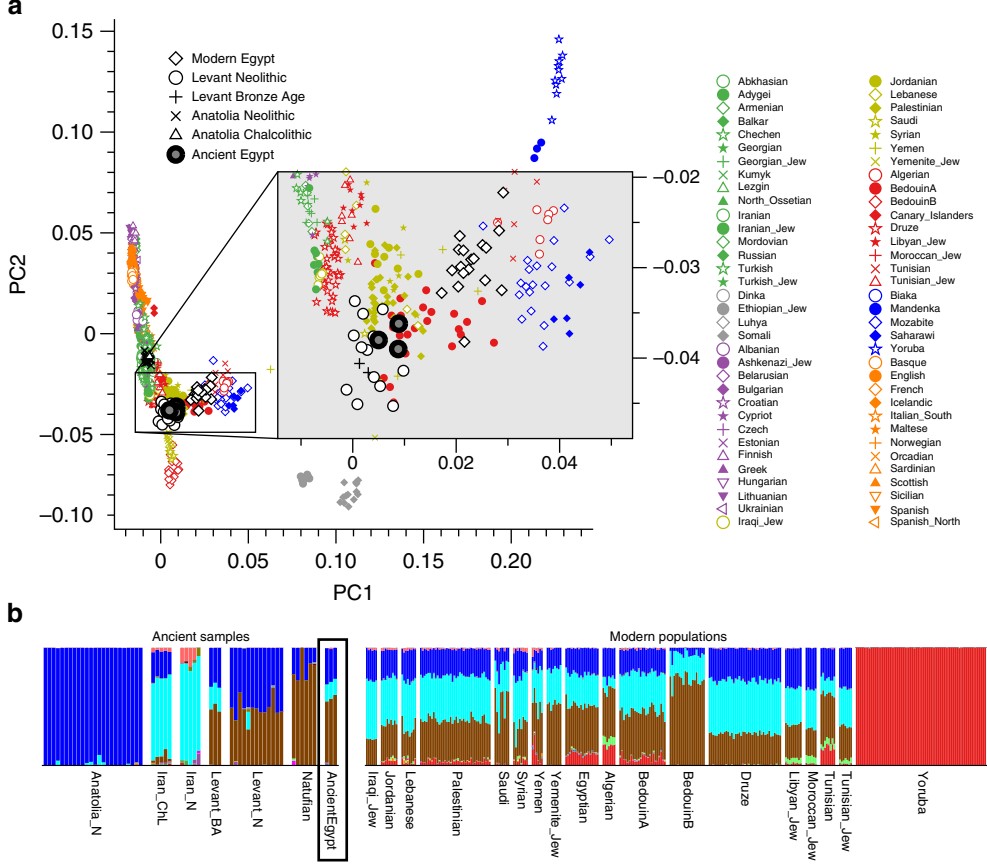

**Figure 4 | Principal component analysis and genetic clustering of genome-wide DNA from three ancient Egyptians. (a)** Principal Component Analysis-based genome-wide SNP data of three ancient Egyptians, 2,367 modern individuals and 294 previously published ancient genomes, **(b)** subset of the full ADMIXTURE analysis (Supplementary Fig. 4).

component (Fig. 5c), confirming the results of the ADMIXTURE analysis. We replicated the results based on $f_3$-statistics using only the least contaminated sample (with <1% contamination estimate) and find very similar results (Supplementary Fig. 5), confirming that the moderate levels of modern DNA contamination in two of our samples did not affect our analyses. Finally, we used two methods to estimate the fractions of sub-Saharan African ancestry in ancient and modern Egyptians. Both qpAdm[35] and the $f_4$-ratio test[39] reveal that modern Egyptians inherit 8% more ancestry from African ancestors than the three ancient Egyptians do, which is also consistent with the ADMIXTURE results discussed above. Absolute estimates of African ancestry using these two methods in the three ancient individuals range from 6 to 15%, and in the modern samples from 14 to 21% depending on method and choice of reference populations (see Supplementary Note 1, Supplementary Fig. 6, Supplementary Tables 5–8). We then used ALDER[40] to estimate the time of a putative pulse-like admixture event, which was estimated to have occurred 24 generations ago (700 years ago), consistent with previous results from Henn and colleagues[16]. While this result by itself does not exclude the possibility of much older and continuous gene flow from African sources, the substantially lower African component in our ~2,000-year-old ancient samples suggests that African gene flow in modern Egyptians occurred indeed predominantly within the last 2,000 years.

**Estimating phenotypes**. Finally, we analysed several functionally relevant SNPs in sample JK2911, which had low contamination and relatively high coverage. This individual had a derived allele at the SLC24A5 locus, which contributes to lighter skin pigmentation and was shown to be at high frequency in Neolithic Anatolia[41], consistent with the ancestral affinity shown above. Other relevant SNPs carry the ancestral allele, including HERC2 and LCT, which suggest dark-coloured eyes and lactose intolerance (Supplementary Table 9).

## Discussion

This study demonstrates that the challenges of ancient DNA work on Egyptian mummies can be overcome with enrichment strategies followed by high-throughput DNA sequencing. The use of ancient DNA can greatly contribute towards a more accurate and refined understanding of Egypt's population history. More specifically, it can supplement and serve as a corrective to archaeological and literary data that are often unevenly distributed across time, space and important constituents of social difference (such as gender and class) as well as modern genetic data from contemporary populations that may not be fully representative of past populations.

The archaeological site Abusir el-Meleq was inhabited from at least 3250BCE until about 700CE and was of great religious significance because of its active cult to Osiris, the god of the dead, which made it an attractive burial site for centuries[2]. Written sources indicate that by the third century BCE Abusir el-Meleq was at the centre of a wider region that comprised the northern part of the Herakleopolites province, and had close ties with the Fayum and the Memphite provinces, involving the transport of wheat, cattle-breeding, bee-keeping and quarrying[42].

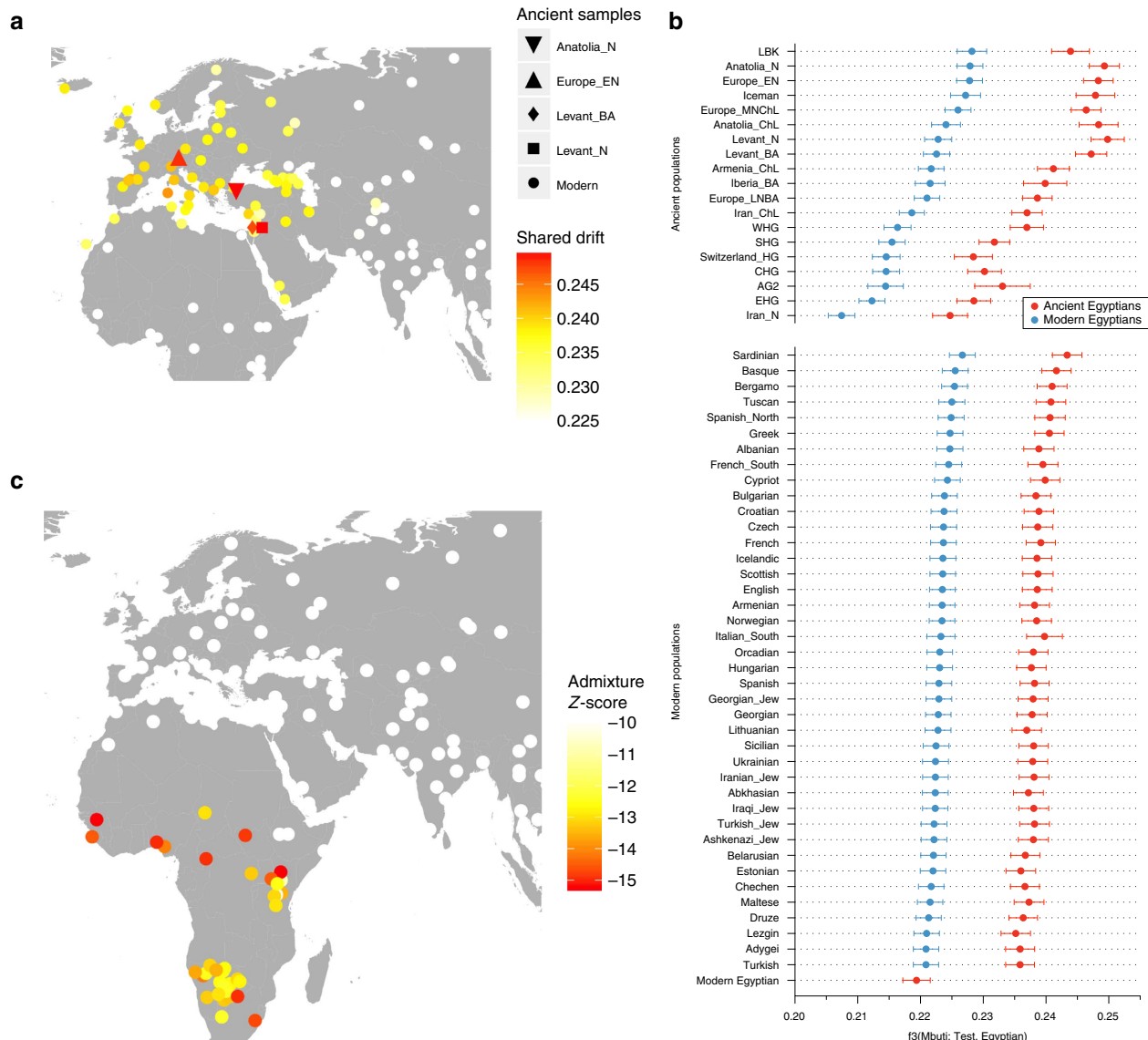

**Figure 5 | Shared drift and mixture analysis of three ancient Egyptians with other modern and ancient populations.** (**a**) Outgroup $f_3$-statistics measuring shared drift of the three ancient Egyptian samples and other modern and ancient populations, (**b**) The data shown in **a**, compared with the same estimates for modern Egyptians, ordered by shared drift with modern Egyptians, (**c**) Admixture $f_3$-statistics, testing whether modern Egyptians are mixed from ancient Egyptians and some other source. The most negative $Z$-scores indicate the most likely source populations.

In the early Roman Period, the site appears to have been the main centre in its own district[42]. Abusir el-Meleq's proximity to, and close ties with, the Fayum are significant in the context of this study as the Fayum in particular saw a substantial growth in its population during the first hundred years of Ptolemaic rule, presumably as a result of Greek immigration[33,43]. Later, in the Roman Period, many veterans of the Roman army—who, initially at least, were not Egyptian but people from disparate cultural backgrounds—settled in the Fayum area after the completion of their service, and formed social relations and intermarried with local populations[44]. Importantly, there is evidence for foreign influence at Abusir el-Meleq. Individuals with Greek, Latin and Hebrew names are known to have lived at the site and several coffins found at the cemetery used Greek portrait image and adapted Greek statue types to suit 'Egyptian' burial practices[2,45]. The site's first excavator, Otto Rubensohn, also found a Greek grave inscription in stone as well as a writing board inscribed in Greek[46]. Taken together with the multitude of Greek papyri that

were written at the site, this evidence strongly suggests that at least some inhabitants of Abusir el-Meleq were literate in, and able to speak, Greek[45]. However, a general issue concerning the site is that several details of the context of the individuals analysed in this study were lost over time. All of the material was excavated by Rubensohn in the early twentieth century, whose main interest was to obtain literary papyri from cartonnage rather than to excavate human remains[47]. As is customary for the time, Rubensohn's archaeological records are highly incomplete and many of the finds made by him were removed undocumented from their contexts. Furthermore, many of his excavation diaries and notes were destroyed during the Second World War[19]. This lack of context greatly diminishes the possibility of 'thick description' of the analysed individuals, at least in terms of their names, titles and materially expressed identity. However, the finds nevertheless hold much promise for a long-term study of population dynamics in ancient Egypt. Abusir el-Meleq is arguably one of the few sites in Egypt, for which such a vast

number of individuals with such an extensive chronological spread are available for ancient DNA analysis. Although we only analysed mummified remains, there is little reason to believe that the burials Rubensohn excavated belonged exclusively to a group of prosperous inhabitants on the basis of the far published references to excavation diaries and Rubensohn's preliminary reports that permit a basic reconstruction. Rather it seems arguable that the complete spectrum of society is represented, ranging from Late Period priests' burials that stand out by virtue of their size and contents to simple inhumations that are buried with little to no grave goods[2]. The widespread mummification treatments in the Ptolemaic and Roman Periods in particular, leading to a decline in standards and costs[48] and the generally modest appearance of many burials further supports this assessment.

By comparing ancient individuals from Abusir el-Meleq with modern Egyptian reference populations, we found an influx of sub-Saharan African ancestry after the Roman Period, which corroborates the findings by Henn and colleagues[16]. Further investigation would be needed to link this influx to particular historic processes. Possible causal factors include increased mobility down the Nile and increased long-distance commerce between sub-Saharan Africa and Egypt[49]. Trans-Saharan slave trade may have been particularly important as it moved between 6 and 7 million sub-Saharan slaves to Northern Africa over a span of some 1,250 years, reaching its high point in the nineteenth century[50]. However, we note that all our genetic data were obtained from a single site in Middle Egypt and may not be representative for all of ancient Egypt. It is possible that populations in the south of Egypt were more closely related to those of Nubia and had a higher sub-Saharan genetic component, in which case the argument for an influx of sub-Saharan ancestries after the Roman Period might only be partially valid and have to be nuanced. Throughout Pharaonic history there was intense interaction between Egypt and Nubia, ranging from trade to conquest and colonialism, and there is compelling evidence for ethnic complexity within households with Egyptian men marrying Nubian women and vice versa[51–53]. Clearly, more genetic studies on ancient human remains from southern Egypt and Sudan are needed before apodictic statements can be made.

The ancient DNA data revealed a high level of affinity between the ancient inhabitants of Abusir el-Meleq and modern populations from the Near East and the Levant. This finding is pertinent in the light of the hypotheses advanced by Pagani and colleagues, who estimated that the average proportion of non-African ancestry in Egyptians was 80% and dated the midpoint of this admixture event to around 750 years ago[17]. Our data seem to indicate close admixture and affinity at a much earlier date, which is unsurprising given the long and complex connections between Egypt and the Middle East. These connections date back to Prehistory and occurred at a variety of scales, including overland and maritime commerce, diplomacy, immigration, invasion and deportation[54]. Especially from the second millennium BCE onwards, there were intense, historically- and archaeologically documented contacts, including the large-scale immigration of Canaanite populations, known as the Hyksos, into Lower Egypt, whose origins lie in the Middle Bronze Age Levant[54].

Our genetic time transect suggests genetic continuity between the Pre-Ptolemaic, Ptolemaic and Roman populations of Abusir el-Meleq, indicating that foreign rule impacted the town's population only to a very limited degree at the genetic level. It is possible that the genetic impact of Greek and Roman immigration was more pronounced in the north-western Delta and the Fayum, where most Greek and Roman settlement concentrated[43,55], or among the higher classes of Egyptian

society[55]. Under Ptolemaic and Roman rule, ethnic descent was crucial to belonging to an elite group and afforded a privileged position in society[55]. Especially in the Roman Period there may have been significant legal and social incentives to marry within one's ethnic group, as individuals with Roman citizenship had to marry other Roman citizens to pass on their citizenship. Such policies are likely to have affected the intermarriage of Romans and non-Romans to a degree[55]. Additional genetic studies on ancient human remains from Egypt are needed with extensive geographical, social and chronological spread in order to expand our current picture in variety, accuracy and detail.

However, our results revise previous scepticism towards the DNA preservation in ancient Egyptian mummies due to climate conditions or mummification procedures[8]. The methodology presented here opens up promising avenues for future genetic research and can greatly contribute towards a more accurate and refined understanding of Egypt's population history.

## Methods

**Ancient DNA extraction and library preparation.** All pre-amplifications steps were carried out in clean room facilities dedicated to ancient DNA work at the University of Tübingen. Before the sampling all samples were UV irradiated for 60 min to reduce modern contamination. In addition, the surface of the bone or tissue samples was removed and the teeth were sampled from inside of the tooth pulp. DNA was extracted from 50 mg bone powder for bone or tooth samples, from 100 mg tissue for soft tissue samples, respectively. A silica purification protocol was applied as described in ref. 56 using the following modifications: the Zymo-Spin V funnels (Zymo Research) were bleached and UV irradiated for 60 min and the total elution volume was raised to 100 µl. Aliquots of 20 µl extract were converted into double-stranded Illumina libraries following a well-established protocol[22] and sample specific barcodes were added to both sides of the fragments via amplification[22,23]. Extraction and library blanks were treated accordingly.

Subsequently, the indexed libraries were amplified using 100 µl reactions for each library containing 5 µl library template, 4 units AccuPrime Taq DNA Polymerase High Fidelity (Invitrogen), 1 unit 10 × AccuPrime buffer (containing dNTPs) and 0.3 µM IS5 and IS6 primers[22], and the following thermal profile: 2-min initial denaturation at 94 °C, followed by 4–17 cycles consisting of 30-s denaturation at 94 °C, a 30-s annealing at 60 °C and a 2-min elongation at 68 °C and a 5-min final elongation at 68 °C. The amplified libraries were then purified using the MinElute PCR purification kit (Qiagen, Hilden, Germany), quantified with Agilent 2100 Bioanalyzer DNA 1000 chips and were used for the enrichment of the human mitochondrial DNA.

For the nuclear capture two additional libraries for selected 40 samples using 20 µl extract were created as described above with the addition of a UDG treatment[27] (see Supplementary Note 2 for details).

**Mitochondrial DNA enrichment and sequencing for sample processing.** All samples were enriched for human mitochondrial DNA via bead capture hybridization as detailed elsewhere[33]. After enrichment the libraries were amplified in 100 µl reactions with 15 µl template, 2 units Phusion High Fidelity DNA polymerase, 1 unit 5 × HF buffer, 0.25 mM dNTPs and 0.3 µM IS5 and IS6 primers[22], and the following thermal profile: 5-min initial denaturation at 95 °C, followed by 16–23 cycles consisting of 30-s denaturation at 95 °C, a 30-s annealing at 60 °C and a 45-s elongation at 72 °C and a 5-min final elongation at 72 °C. Subsequently, the libraries were purified and quantified as described before and paired-end dual index sequencing was carried out on an Illumina HiSeq 2500 platform by 2 × 100 + 7 + 7 cycles following the manufacturer's protocols for multiplex sequencing (TruSeq PE Cluster Kit v3-cBot-HS).

**Mitochondrial DNA sequence processing and alignment.** The resulting FastQ files have been processed using EAGER v1.92 (ref. 57). To achieve improved coverages at both ends of the mitochondrial reference, we used the CircularMapper option in EAGER. All reads with a mapping quality of at least 30 were kept for the subsequent analysis. Duplicate reads have been removed using DeDup v0.9.10, included in the EAGER pipeline. The coverage and statistics calculation has been performed inside the EAGER pipeline and indels have been realigned using RealignerTargetCreator and IndelRealigner from the GATK[58]. Mitochondrial haplogroups have been determined using HaploGrep 2 (ref. 59). Further details of the analysis parameters can be found in Supplementary Note 3. As can be seen in Supplementary Data 1, we achieved coverages ranging from 11-fold up to 4284-fold on the mitochondrial genome, with an average of 408-fold.

**Mitochondrial DNA authentication and contamination assessment.** Accompanying measures to limit contamination of the libraries in the laboratory work, in silico analysis has been done in order to authenticate samples and further

determine the amount of potential contamination on the mitochondrial level. Negative controls were processed in parallel with samples. The former show no substantial mapping rates and suggest that the amount of DNA introduced during laboratory work could be kept on a minimal level. The authenticity of the samples has been further assessed by applying a number of methods and criteria. MapDamage 2.0 (ref. 60) has been used to evaluate fragment lengths and nucleotide misincorporation patterns of the provided samples, all of which showed levels that are characteristic for ancient DNA[13]. The degree of mitochondrial DNA contamination as well as contamination estimates based on the deamination patterns have been assessed using schmutzi[15], generating consensus sequences of both contaminant and supposedly endogenous DNA simultaneously. Furthermore, only samples with less than 3% estimated contamination based on deamination and degree of mitochondrial contamination have been used for further downstream analysis. We furthermore determined whether there are inconsistencies between our haplogroup assignments of the mitochondrial and the nuclear capture respectively, but did not find any (see Supplementary Data 3 for details). As can be seen in Supplementary Table 1, our samples showed damage on both 3′ and 5′ ends of reads in the range of 5% up to 49%, with an average of 14%. Furthermore, the contamination estimation methods showed very low levels of contamination after comparison to a database of putative contaminants, as provided by the used method schmutzi. For all samples, the observed contamination estimates prove to be less than our defined threshold of 3%, except for three samples (JK2879, JK2883, JK2896) where a visual inspection of sequence assemblies was done as described in Posth et al.[61] to identify potential contaminating lineages and ensure consistency of the generated consensus mitochondrial genome. As an additional measure, we used the built-in feature 'log2fasta' of the tool schmutzi to only incorporate bases in our final consensus sequence with a significant likelihood to be non-contaminated as defined by the method itself. In order to do this, we applied several quality thresholds ($q = 0, 20, 40, 80$) in our analysis and used a moderate filtering value that did not change our consensus sequence to undefined positions to a larger extent. We ultimately chose a value of $q = 20$ for filtering with 'log2fasta', but even more strict filtering with $q = 40$ preserved our haplotyping calls to be consistent. However, filtering even stricter introduced more undefined positions ('N') due to missing support, potentially hindering sequence-based analysis more dramatically than our frequency-based analysis, which is why we kept a quality threshold of $q = 20$, following cutoffs that other authors have been using, too[61].

**Nuclear DNA capture.** The non-UDG and UDG treated libraries were enriched by hybridization to probes targeting approximately 1.24 million genomic SNPs as described previously[25]. The target SNPs consist of panels 1 and 2 as described in Mathieson et al.[41] and Fu et al.[26] (see Supplementary Note 2 for details).

For each of the 40 samples, we sequenced two captured libraries: one with enzymatic damage repair (UDG), one without (non-UDG). For all samples, we used the EAGER pipeline version 1.92.15 (ref. 57), with default parameters, and with the option to keep only merged reads. We determined the sex of each sample by obtaining the average coverage on X chromosome, Y chromosome and autosomal SNPs in the capture pool using a custom script. We flagged samples as 'male' when the ratio of X and autosomal coverage was lower or equal than 0.75 and the ratio of Y and autosomal coverage was greater or equal than 0.25. We flagged samples as 'female' when the ratio of X and autosomal coverage was greater than 0.75 and the ratio of Y and autosomal coverage was lower than 0.25. For all male samples that had at least a total number of 150 SNPs on chromosome X covered twice, we obtained contamination estimates using the ANGSD software[62], using the 'MoM' estimate from 'Method 1' and the 'new_llh' likelihood computation. Supplementary Data 2 summarizes all these results. In some cases, ANGSD finished with an error, as indicated in the table. Entries with 'n/a' are either female or have insufficient coverage on the X chromosome.

Three samples were selected for down-stream analysis: JK2134, JK2888 and JK2911. In all three of these samples, contamination estimates are acceptable, and similar in both UDG and non-UDG libraries as can be seen in Supplementary Data 2. Furthermore, in all three samples the non-UDG library showed DNA damage over 8% in the first base pair of reads, which is within the expected range of damage for ancient DNA of this age.

**Nuclear data analysis: genotyping.** We called genotypes from the UDG treated data for the three individuals by sampling a random read per SNP in the SNP-capture panel, using a custom tool 'pileupCaller', available at https://github.com/stschiff/sequenceTools. The resulting genotypes were merged with data from two other data sets: First, 2,367 modern individuals genotyped on the Affymetrix Human Origins Array[34,35]; second, 294 ancient genomes[36].

**Nuclear data analysis: ADMIXTURE.** We used the ADMIXTURE software on the merged data set to cluster ancestry proportions using different numbers of clusters[37]. The lowest cross-validation error was obtained using $K = 16$ and we show the results of that run in Supplementary Fig. 4. A subset is shown in Fig. 4b.

**Nuclear data analysis: PCA.** We performed PCA on the joined data set using the 'smartpca' software from the Eigensoft package[63]. For the plot shown in

Supplementary Fig. 3, we used a selected set of European populations as shown in Supplementary Note 2.

**Nuclear data analysis: $f_3$-statistics.** We used the 'qp3pop' tool from the Admixtools package[39] to compute Outgroup $f_3$-statistics of the form $f_3$(Mbuti; Egyptian, X), where 'Egyptian' means either ancient and modern Egyptian, and 'X' runs over all populations in the merged data set. For the plot in Fig. 5b, we ordered all results based on the result using the modern Egyptian samples and show the top hits. For the map plot in Fig. 5a we placed all modern populations on their sampling locations obtained from Lazaridis et al.[34], and added selected ancient populations that stood out from the background, as shown in Figure 5b. We then used the 'qp3pop' tool to compute $f_3$-statistics of the form $f_3$(Egyptian; Ancient Egypt, X), where X runs over all populations in the merged data set. Fig. 5c shows a similar plot as in Fig. 5a, but with the colour code indicating the Z score for this latter $f_3$-statistics, where a negative Z score indicates a probable source for admixture.

Since two of the three selected samples had contamination rate estimates over 5%, we repeated this analysis using only sample JK2911, which has the highest SNP coverage and a contamination estimate of below 1%. The result is shown in Supplementary Fig. 5, with very similar results as when using all three samples, indicating no effect of contamination on our results.

**Sequence-based mitochondrial analysis.** In order to detect genetic similarities or distances between our three ancient Egyptian populations ($n = 90$) and present-day populations (see Supplementary Note 4), we collated a data set of Egyptian ($n = 135$) and Ethiopian ($n = 120$) mtDNA sequences from the literature for the respective area in upper Egypt, the El-Hayez oasis[30] and Ethiopia[17]. We calculated genetic distances ($F_{ST}$) based on the full mtDNA of these individuals. $F_{ST}$ values were calculated using Arlequin v3.5.2.2 (ref. 28), applying the Tamura and Nei substitution model[64] and a respective gamma value of 0.260. To determine the most suitable parameter set and substitution model, we used jModelTest v2.1.10 (ref. 65) and selected the parameters suggested by the Akaike and Bayesian information criterion (AIC and BIC). P values for the calculated $F_{ST}$ values were corrected for multiple comparisons to minimize the probability of type I errors (false positives) using the Benjamini–Hochberg method[66], a false discovery rate-based method implemented in the *p.adjust* function in R 3.2.3 (The R Project for Statistical Computing 2011, https://www.r-project.org/). We split our individuals in three groups (Pre-Ptolemaic, Ptolemaic and Roman Period) based on the [14]C dates obtained from the samples (Supplementary Data 1). However, as the intra-group distances of our three ancient populations were not significantly different from each other, we merged all three ancient populations in a single set to perform $F_{ST}$ analyses between modern populations and the ancient meta population with more statistical power than keeping the individual populations separate. Our results can be found in Supplementary Table 2.

**Sequence-based mitochondrial analysis: multidimensional scaling (MDS) analysis.** To determine the relationships between our ancient samples from the Pre-Ptolemaic, Ptolemaic and Roman time periods in contrast to modern populations in the respective areas, we performed a multi-dimensional scaling (MDS) analysis of the HVR-1 sequences (Supplementary Data 4).

The genetic distances were calculated in Arlequin v3.5.2.2 using the Tamura and Nei substitution model and a gamma shape value of 0.26, determined to be the best setting for the data using jModelTest v2.1.10. We selected the best parameters suggested by the Akaike and Bayesian information criterion (AIC and BIC).

We used the linearized Slatkin's $F_{ST}$ values[67] based on our data set of HVR-1 sequences and visualized the calculated $F_{ST}$ values in a two-dimensional MDS plot with GNU R 3.2.4 using customized R script embedded in the *vegan* package (Fig. 3c). Our ancient Egyptian samples have been pooled here in order to provide more significant statistical evidence in the analysis, which can be justified due to their relatively small intraspecific differences between our three investigated time periods in the previous $F_{st}$ analysis on their full mitochondrial genomes. The closest populations on the MDS with respect to our ancient meta population (AEGY) are modern populations from Saudi-Arabia, Kuwait, the United Arab Emirates, Yemen and other Near-East populations, whereas the individuals from another ancient population from Turkey (TRO) show more relatedness to modern North-African and populations from the Levantine. For details on the geographic mapping, see Supplementary Note 4.

**Sequence-based mitochondrial analysis: effective population size estimation using BEAST.** We used the 90 mitochondrial genomes obtained in this study, together with 135 modern Egyptian mtDNA genomes from Pagani and colleagues[17] and Kujanova and colleagues[30] for Bayesian reconstruction of population size changes through time. We partitioned the alignment using the krmeans algorithm in PartitionFinder2 (ref. 68) with a search through all models available excluding I + G models as it has been argued that gamma-invariable models are not biologically meaningful for data sampled at intraspecies level[69]. The BIC best-fit partitions (three partitions: 7212, 2367 and 6999 nt, assigned TRN, K81uf + I and TRN + I, respectively, as the best model) were used for BEAST v 1.8.3 analysis[32] with unlinked site and clock models and linked tree model.

We used averages from the calibrated radiocarbon age ranges for each ancient sample as tip dates for molecular clock calibration. We conducted Bayesian inference using strict clock with an uninformative CTMC reference prior for each partition and Bayesian SkyGrid tree prior with 50 parameters (gamma prior with shape 0.001 and scale 1,000). MCMC chain was run for 300 million steps with sampling every 30,000th step and initial 10% discarded as burn-in. We inspected mixing and convergence in Tracer v 1.6 (ref. 70). Effective sample size for all parameters exceeded 100.

The obtained Bayesian SkyGrid plot indicates a fairly stable slightly decreasing effective population size for the studied population over the last 5,000 years (Fig. 3d and Supplementary Fig. 2). The average median population size over the sampled ancient period, expressed as female effective population size times generation time, was estimated to 1,625,187 (95% HPD 693,670–4,490,725), which assuming generation time of 29 years and equal male and female effective population size rescales to 112,082 (95% HPD 47,839–309,705) individuals (see Supplementary Table 4).

**Frequency-based mitochondrial analysis: principal component analysis.** We performed a PCA to define relationships between our three ancient Egyptian populations based on their haplogroup compositions and modern, present-day populations from Europe, the Near East, West Asia and Africa. To generate the PCA, we divided our haplogroups in the following 20 groups: H, HV, I, J, K, L0, L1, L2, L3, L4, M1, N, R, R0, T, T1, T2, U, W, X and all remaining other haplogroups (see Supplementary Data 1 for haplogroups). Subsequently, we generated a table of the respective intra-population frequencies. The PCA was performed using the *prcomp* function for categorical PCA implemented in GNU R 3.2.4 and plotted in a two-dimensional space, displaying the first and second principal components and shown in Fig. 3b.

**Frequency-based mitochondrial analysis: test of population continuity (TPC).** Our intent was to determine whether we can detect traces of genetic continuity between our three ancient populations and two comparative modern data sets. The applied method was first used and described by Brandt et al.[29]. We generated counts of 22 haplogroups determined manually to be most descriptive for our three ancient populations and chose a set of priors for effective population size, generation length and furthermore evaluated further parameters (see Supplementary Note 5). Especially since we are unable to determine a real value of population size during this time period, we relied on historic records for the Fayum oasis and estimated a conservative population size from this (Supplementary Table 4). To even further ensure that these chosen values are not changing our results drastically, we evaluated ranges around these assumptions to test whether our results changed significantly.

**Y-chromosomal and phenotypic analysis.** We determined the Y chromosomal haplogroups for our three nuclear captured individuals by examining the state of phylogenetic relevant SNPs present in ISOGG version 11.228 (accessed 19 August 2016). The assignment was performed with reads that show a mapping quality of more than 30 only. We derived the haplogroups by identifying the most derived Y chromosomal SNPs in each individual (see Supplementary Table 3 for details).

Our analysis furthermore shows that derived alleles for the genes SLC24A5, known to be responsible for partially lighter skin pigmentation were present in both JK2888 and JK2911 (see Supplementary Note 6 for details). For further genes such as SLC45A2, LCT and EDAR we were unable to find derived alleles for both JK2888 and JK2911. For JK2134, there was no sufficient coverage after quality filtering at all the specific sites, which is why the analysis revealed no further clues.

**Data availability.** The mapped BAM files for the 90 mitochondrial samples and three nuclear samples are deposited in the European Nucleotide Archive (http://www.ebi.ac.uk/ena) with the study ID ERP017224.

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

## Acknowledgements

We thank Gabriel Renaud for the help with the contamination estimation; Guido Brandt and Anna Szécsényi-Nagy for sharing the mtDNA database; Annette Günzel for designing the publication graphics for Fig. 1; Kerttu Majander for drawing icons and images for Fig. 3d of the publication; Sarah Inskip for comments on the manuscript; Claus D. Claussen for his support with scanning the mummified heads. M.M. was supported by the Foundation for Polish Science. K.H. is supported by the Deutsche Forschungsgemeinschaft (DFG FOR 2237).

## Authors contributions

V.J.S., W.H., S.S. and J.K. designed the experiments. K.H. and B.T. provided samples for analysis. V.J.S, B.W., C.U. and M.F. performed the skeletal sampling. V.J.S., A.F., C.U. and E.R. performed the ancient DNA experiments. A.P., S.S., W.H., M.M., C.-C.W. and K.N. analysed the data. V.J.S., A.P., W.P.vP., W.H., S.S. and J.K. wrote the manuscript with contributions from all co-authors. All authors read and approved the manuscript.

## Additional information

**Competing interests:** The authors declare no competing financial interests.

