## [Peer Review File · Nature Communications]

Reviewers' comments:

Reviewer #1 (Remarks to the Author):

This is an interesting paper that presents important insights into the genetic history of the ancient Egyptian population. To my knowledge, this work represents the first detailed population genetic study on ancient Egyptian mummified material, by the reconstruction of 91 mitochondrial genomes and three genome-wide datasets. It has to be acknowledged that the authors applied strict quality controls and thereby managed to unequivocally prove the authenticity of their findings. Based on this data, the long-lasting and often dogmatic discussion on whether ancient DNA could survive in ancient Egyptian mummies can be finally put to rest.

Both, the molecular approach and the data analysis was very well done and the results provide new and important insights into the continuity of the population of the ancient Egyptian site and the shared ancestry of the ancient Egyptians with Near Easterners. The work represents an important contribution to the study of ancient human history in Egypt.

Nevertheless, the work could be further improved by providing more details on the investigated mummies and a more detailed analysis of a possible foreign influence at the site. It would be interesting to understand if the methodological approach could allow to identify single individuals that have a different origin, such as Ptolemaic or Roman, and what it would tell us about burial practice and mummification techniques.

In the following, I have listed some specific comments:

1. I would have liked to see some comments on the debate about DNA preservation in ancient Egypt and in particular the mentioning of some previous work. I think it would strengthen the impact of this paper, if the authors would underline the importance of their findings with regard to the general skeptics on the presence of ancient DNA in hot and arid climates.
2. I am particularly surprised that the authors have not cited the previous work on the material that derives from the same collection. As far as I understand, Khairat et al. (J. Appl. Genet., 2013) and Lalremruata et al. (Plos One, 2013), also used samples from the mummy collection at the University of Tuebingen. As the authors of this work have used some radiocarbon dates from Lalremruata et al. (see Suppl.), I assume that the samples were taken from the same collection. It should be explained why the previous work was not mentioned.
3. Lines 55-57: "However, methodological problems and contamination obstacles... hampered direct investigations...". What exactly were these problems and obstacles?? I assume that a major part of the problem was the disbelief of many scholars that ancient DNA could survive in Egyptian mummies due to the hot climate. What were the other reasons? Why has it become possible now? (see also comment 1)
4. Lines 66-72. Evidence for foreign influence: Is this true for the studied samples? What do we know about the mummies regarding names, burial practice, coffin styles, etc.? This is an interesting point! Why didn't the authors use their data to prove or disprove this assumption? The authors should mention whether they tried to see any foreign influence and how could this have influence their analysis!
5. Lines 75-77. "In particular, the site holds much promise for studying changes in its population structure from the late Dynastic Period to the present day." Why is this the case? Is it due to the

better DNA preservation in the later mummies?? Please explain!

6. Line 78. 151 mummified remains: It would have been interesting to learn more about these mummies. Is there any information on the age and sex of the mummies? How were they mummified? How is the state of preservation? How many of them have an "identity" (e.g. name inscription)? (see also comment 4).

7. Line 92. Yields of preserved DNA: What's about the preservation of non-human DNA? Did the authors have a look into the metagenomics data? It is surprising to see that the authors, who published a series of articles on ancient pathogen DNA, didn't check for the presence of diseases such as TB, malaria or toxoplasmosis, which were all reported in previous studies. Please comment.

8. Lines 98-99. DNA damage in mummies: Is this true for all mummies or just Egyptian mummies? Is this influenced by the mummification process? Could it be different in natural mummies?

9. Lines 137-140. "The affinity to the Middle East finds further support by the Y-chromosome haplogroups..." This is true, but the two haplogroups are believed to have different origins (J Western Asia, E1b1b1 North Africa). Moreover, both individuals with haplogroup J are from the pre-ptolemaic period and the individual with haplogroup E from the Ptolemaic period. Does this tell us something about their origin or any differences? The authors should comment on this.

10. Line 144. "The finding of a continuous population..." Is it really a continuous population, or are the foreign influences in the Ptolemaic-Roman period probably too little to be accounted?? Would it be possible at all to trace down foreigners with this approach?? Please comment!

11. Conclusions. What are the particular challenges of working with ancient Egyptian mummies? (see comment 3.). It seems that there was some work published on ancient pathogen DNA in Egypt, but very little on human DNA. What is the reason? Was it mainly a methodological problem, such as the non-availability of high-throughput sequencing and enrichment or are there particular issues with contamination or inhibition? I think it could be interesting for the reader to understand the potential and obstacles of working with Egyptian mummies.

12. Conclusions. The authors should discuss the possible influence and presence of foreigners at the archaeological site Abusir el-Meleq and how their approach could be used to detect it. Although there appears to be a continuity in the overall population, there could be maybe some single individuals that originate from the Ptolemaic or Roman population. Would it be possible to identify those people and could this help to understand the use of certain mummification practices?

13. Conclusions. I agree that in other sites, especially in the south of Egypt, there could be a much higher influence of Sub-Saharan populations. It is well known that there were close trading connections to Nubia and other Sub-Saharan areas during the Middle and New Kingdom. What does this mean for the interpretation of your findings?

Reviewer #2 (Remarks to the Author):

This paper is written like similarly structured ancient DNA papers, with a focus on the production of the dataset, a set of standard analyses performed for small sample sizes and extrapolation to prehistorical conclusions. The problem with this approach is that for the majority of studies, prior genetic analysis (and archaeological etc) has already been performed. The ancient DNA papers largely

ignore the other fields and are written 'hypothesis-free.' Their scientific value becomes limited as each new paper attempts to devise and answer hypotheses that are often longstanding, involving significant repetition. This paper should and could be re-written to answer a question. It would start with a review of prior historical and genetic work performed on Egyptian populations, narrow down to 1-2 specific hypotheses (Can we confirm that slave trade into Egypt and Near East occurring in the past XX centuries changed the genetic composition of Egyptians represented by urban cohorts in Cairo?) Is this a paper about success in sequencing ancient Egyptians or about resolving the population history of Egyptians given new mtDNA data?

Unfortunately, the authors appear to have ignored several earlier papers on Egyptians (see below for 3 examples, there may be others). Their primary result, stated in the abstract as "Our analyses reveal that ancient Egyptians shared more ancestry with Near Easterners than present-day Egyptians, who received additional Sub-Saharan admixture in more recent times" was shown in 2012 using genome-wide SNP array data from Egyptians (Henn et al., PLoS Genetics: Table 1). That is, the sub-Saharan African ancestry is of recent origin in Egyptians dating to AD 1250 (~24 generations ago). Indeed those authors argued that the source of this sub-Saharan ancestry is more likely to be Nilotic than West African as modeled in the current analyses. This is especially problematic as it is the only result reported in the Abstract itself. And again in the conclusion: "By comparing ancient with modern Egyptians, we remarkably found an influx of additional Sub-Saharan African ancestry after the Roman Period, which is visible at both mitochondrial and autosomal level." This result is not remarkable, it is what is expected given prior published analyses.

For example, Kujanova et al. (2009) analysis of mtDNA and Y-chromosomal data from an Egyptian oasis isolate from middle Egypt is not cited. They found that sub-Saharan mtDNA L-lineages constituted 30% of their dataset, indicating a strong sub-Saharan component on the maternal side, which was largely absent on the Y (6% M2-derived lineages). Kujanova et al. also argued for a recent migration event to account for the L lineages: "The absence of any signs of local accumulation of diversity in the L haplotypes in el-Hayez seems to favor the idea of these lineages being recent introductions into the Egyptian Western Desert."

And in Pagani et al. (2015, AJHG) show that "the average proportion of non-African ancestry in the Egyptians to be 80% and dated the midpoint of the admixture event by using ALDER to around 750 years ago (Table S2), consistent with the Islamic expansion and dates reported previously."

In contrasting this paper with another similar high-profile journal paper, Pagani et al. (2015) AJHG, I do not find the results in this manuscript to be major scientific contribution and would be more appropriate for a specialized journal.

Section beginning line 144: The authors report apparent congruity between their population sizes estimates and those from the Fayum in the Ptoleimaic Period with a 95% credible interval range between 50,000 - 280,000. 1) This is the N_e estimated from mtDNA. In order to compare with the true N_e it need to be scaled 4-fold. 2) The Y-axis scale on Figure 3D does not match the results reported in the text. Indeed the N_e appears to be between 1-2 million durin the Ptolemaic period (black line). 3) Assuming even that the numbers in the text are correct, (115,000 50th estimate) the scaling for females vs. female+male as reported in the historical record would indicate a large divergence between the genetic and historical estimates. 4) No prior on the mutation rate, which will strong influence N_e , is given in the supplement. This is the appropriate place to integrate assumption of g (generation time)

Line 197: Lazaridis et al. (2014) contained 18 Egyptian sample of uncertain provenance (Cairo-urban area). Other autosomal SNP array datasets at least including many more individuals are not used here.

Contrasting 3 individuals to 18 contemporary ones in order to indicate extrapolate evidence for a pattern that "6 and 7 million Sub-Saharan slaves to Northern Africa over a span of some 1250 years" seems a strong extrapolation on its own. As the authors admit.

Line 180: "This individual had a derived allele at the SLC24A5 locus, which encodes for light skin pigmentation..." SLC24A5 is among at least 20 skin pigmentation genes known to contribute to melanin variability in contemporary human populations. It alone does not 'encode' for light skin pigmentation. Indeed, many individuals in Cape Verde, Africa carry this allele (which does lighten pigmentation, accounting for 7% of the variance in pigmentation) but their overall phenotype would be still be twice as dark as an average European (Beleza et al. 2013, PLoS Genetics).

Note 8: Test of Population Continuity: the analysis here was not described. Other than collapsing mtDNA lineages into haplogroup frequencies to compare ancient and contemporary groups, there is no description of what the actual test was. Even if the method was described in Brandt et al. (2013) [not even in the main text, only in their supplement], the authors should lay out the assumptions, parameter choices and models invoked in using this method. Why for example, is TPC preferable over Approximate Bayesian Computation models typically used to test the relative likelihood of two different population demographies (in this case continuity w/ minimal drift vs. migration).

Reviewer #3 (Remarks to the Author):

In this very well written, concise yet informative manuscript, Verena Schuenemann and colleagues report genetic data obtained from ancient Egyptian mummies. Because DNA is not very well preserved in mummies, the authors used capture techniques to enrich their libraries for specific human target regions. This technique resulted in high-coverage mitochondrial genomes for 91 samples and an additional set of autosomal SNPs for 3 of those samples. When comparing to modern data, the authors find convincing evidence of an influx of Sub-Saharan alleles in more recent times.

To obtain these results, the authors used by now well established analysis techniques, which were applied carefully to the degree I can judge. The analyses does appear sound and I do not have any concern that the observed genetic differences between ancient and modern Egyptians is real.

I'm less confident that this result is surprising given the archaeological and historical knowledge of the region. I'm not a specialist of that history, but I would have enjoyed reading this paper much more if the introduction and discussion / conclusion would relate this finding better with the existing literature on the Egyptian history. Just stating that "Egypt provides a privileged setting for studying population history" fails a bit short of discussing open questions the study of Egyptian mummies might help to settle. Indeed, the way the paper is pitched know the real emphasis is just put on the fact that DNA could be obtained from mummies.

Another aspect that I feel is missing is a brief discussion on how the bias from only studying individuals that were mummified (as compared to random individuals from the ancient Egypt) might have let to the lack of a Sub-Saharan component in the ancient individuals.

I just found one type: L123: three (instead of tree)

REVIEWERS' COMMENTS:

Reviewer #1 (Remarks to the Author):

The authors have taken into consideration all my comments and the revised manuscript has improved significantly.

However, I do not fully agree with the response to my first query and the way the introduction was modified accordingly (lines 84-105). Although it is true that most of the previous work was done in the "PCR era of ancient DNA", it does not necessarily mean that the previous findings were mainly the result of modern contamination. In particular, as most of the criticism was based on a general assumption that DNA may not survive in hot and dry climates. In this work, the authors have successfully demonstrated otherwise and they should therefore come to the conclusion that the general doubts on preservation of ancient DNA in Egyptian mummies (e.g. citations 8,9) were premature.

It is true and highly appreciated that the authors performed the first in-depth ancient DNA analysis using high-throughput sequencing methods, so there is no need to generally doubt previous work that was done on Egyptian mummies.

I would once more recommend that the authors present the previous work in a more balanced way, mentioning the issues with the PCR based approach, but also acknowledge that DNA seems to survive in ancient Egyptian mummies despite previous skepticism.

If the authors are willing to address this issue, I would be happy to accept the manuscript for publication.

Reviewer #3 (Remarks to the Author):

This is now the second time I review this manuscript and I'm pleased to see that the manuscript has greatly improved from its initial version. In particular, the presented research is now appropriately embedded in the general historical context of the region and the authors now mention the specific questions or hypothesis they test more explicitly. I also remain happy with the analyses presented. It is indeed questionable how accurate ancient admixture rates can be learned from either a single locus (mtDNA) or just three samples in a case in which admixture is so recent that ancestry blocks are necessarily large and the variation in admixture rates between individuals expected to be very high. However, the authors are aware and transparent about these shortcomings in their data. Yet, as a result, the presented results do still not go much beyond confirming some previously noted findings and the real contribution of the work seems to be the success in obtaining reliable genetic data from Egyptian mummies. The importance of that finding to the relevant community is something I can not judge.

I just stumbled across one typo: on L164: repetition of "still".

"Ancient Egyptian mummy genomes suggest an increase of Sub-Saharan African ancestry in post-Roman periods" by Verena J. Schuenemann and colleagues

Response to reviewer comments:

Please find below a point by point response to the comments made by each of the three reviewers. Their individual comments are in black, the matching response in blue.

Reviewer #1:

Q1. I would have liked to see some comments on the debate about DNA preservation in ancient Egypt and in particular the mentioning of some previous work. I think it would strengthen the impact of this paper, if the authors would underline the importance of their findings with regard to the general skeptics on the presence of ancient DNA in hot and arid climates.

Answer:

We have substantially rewritten the introduction (lines: 84-105) to address results (or lack thereof) from previous studies and questions about the feasibility to retrieve ancient human DNA from samples from hot and arid climates.

Q2. I am particularly surprised that the authors have not cited the previous work on the material that derives from the same collection. As far as I understand, Khairat et al. (J. Appl. Genet., 2013) and Lalremruata et al. (Plos One, 2013), also used samples from the mummy collection at the University of Tuebingen. As the authors of this work have used some radiocarbon dates from Lalremruata et al. (see Suppl.), I assume that the samples were taken from the same collection. It should be explained why the previous work was not mentioned.

Answer:

It is correct that we used 5 radiocarbon dates reported in Lalremruate et al. 2013 and generated 85 new ones. The study Khairat et al. 2013 is now cited in our revised introduction: lines 103-105. All studies on the Tuebingen mummy collection (Lalremruate et al. 2013, Khairat et al. 2013, Nicholson et al. 2011 and Welte 2016) are now cited in our revised sample description: lines 173-175.

Q3. Lines 55-57: "However, methodological problems and contamination obstacles... hampered direct investigations...". What exactly were these problems and obstacles?? I assume that a major part of the problem was the disbelief of many scholars that ancient DNA could survive in

Egyptian mummies due to the hot climate. What were the other reasons? Why has it become possible now? (see also comment 1)

Answer:

This is not a matter of beliefs, but based on the fact that many/most of the early studies did not withstand the scrutiny of ancient DNA authentication criteria. These are particularly important when reporting results from challenging climates (see e.g. Bollongino et al. 2008, Comptes Rendus Palevol 7(2):91-98). Many of the studies of Egyptian remains were conducted in the classical 'PCR era of ancient DNA', a technique which favors intact DNA molecules and is thus prone to modern human DNA contamination. It furthermore does not allow to study DNA damage patterns, that were found to be among the most reliable criteria for ancient DNA authenticity (Stoneking & Krause 2011). It is well known that the length of ancient DNA molecules is around 50bp and the overall amount of DNA damage roughly correlates with (thermal) age (Matthew Collins, Susanna Sawyer etc...), which makes STR profiling (up to 350 bp!) rather difficult. NGS techniques have the advantage to utilise the large number of independent DNA fragments (usually not accessible to PCR) in order to study contamination. Further, many previous studies have used muscle tissue, which we clearly show to be the least promising/appropriate material when studying ancient DNA. We expanded our discussion on preservation conditions and previous work in the introduction (lines: 84-105), however much of this debate can be found in Lorenzen et al. 2010 and was not repeated here.

Q4. Lines 66-72. Evidence for foreign influence: Is this true for the studied samples? What do we know about the mummies regarding names, burial practice, coffin styles, etc.? This is an interesting point! Why didn't the authors use their data to proof or disproof this assumption? The authors should mention whether they tried to see any foreign influence and how could this have influence their analysis!

Answer:

We have extensively rewritten the introduction and sample information to include all available contextual information on the individuals under study (lines: 111-137, 162-172). Unfortunately, this information is scarce and not sufficient to formulate strong hypotheses that could be tested in a formal manner. This issue is now detailed in our discussion: lines 361- 371. We do, however, fully address the reliability of our sample as being representative of the community at the time (see discussion: lines: 371-385).

Q5. Lines 75-77. “In particular, the site holds much promise for studying changes in its population structure from the late Dynastic Period to the present day.” Why is this the case? Is it due to the better DNA preservation in the later mummies?? Please explain!

Answer:

Unfortunately, mummies from the Old till early New Kingdom are not present at the site or and not included in our data set, which focusses on the three consecutive periods. The site is mainly occupied during the Late Period till Roman times according to written sources, and thus would allow the study of an extended temporal transect. We furthermore find in more than 50% of all remains authentic ancient DNA preserved, suggesting this to be an ideal site for further studies. We removed the sentence and included a more intensive introduction and discussion of the site, see Q1 and Q4.

Q6. Line 78. 151 mummified remains: It would have been interesting to learn more about these mummies. Is there any information on the age and sex of the mummies? How were they mummified? How is the state of preservation? How many of them have an “identity” (e.g. name inscription)? (see also comment 4).

Answer:

As mentioned in response to Q4, the individual information is scarce. We have included all available information and anthropological data (see lines: 176-192) in the revised manuscript and supplementary information now.

Q7. Line 92. Yields of preserved DNA: What’s about the preservation of non-human DNA? Did the authors have a look into the metagenomics data? It is surprising to see that the authors, who published a series of articles on ancient pathogen DNA, didn’t check for the presence of diseases such as TB, malaria or toxoplasmosis, which were all reported in previous studies. Please comment.

Answer:

We used our latest metagenomic pipeline (Herbig et al. 2016 biorxiv) in order to find evidence for authentic ancient pathogens, we however failed to find any clear signal for the pathogens reported in previous studies. More analyses on teeth and individuals with clear pathological lesions might be more promising for future research on ancient pathogens from Egyptian mummies.

Q8. Lines 98-99. DNA damage in mummies: Is this true for all mummies or just Egyptian mummies? Is this influenced by the mummification process? Could it be different in natural mummies?

Answer:

To our knowledge, there is no comprehensive and comparative survey on mummies from a global variety published to date. For natural mummies, such as for example from Peru (dry mummies) or the Tyrolean iceman (ice mummy), there seems to be no general pattern observable other than those from high altitudes, i.e. 'perma-frost-conditions, show higher yields of authentic DNA and reduced damage patterns (e.g. Iceman, Saqqaq).

To our knowledge, no systematic tests have yet been done on mummies, as for example for bones and eggshells in Allentoft et al (2012). There have been attempts to simulate DNA decay by burying material and investigating after several years how much DNA damage was introduced, these results were however inconclusive.

Q9. Lines 137-140. "The affinity to the Middle East finds further support by the Y- chromosome haplogroups..." This is true, but the two haplogroups are believed to have different origins (J Western Asia, E1b1b1 North Africa). Moreover, both individuals with haplogroup J are from the pre-ptolemaic period and the individual with haplogroup E from the Ptolemaic period. Does this tell us something about their origin or any differences? The authors should comment on this.

Answer: (to be included in the manuscript and referenced here)

We agree with the reviewer that Y chromosomal haplogroup J is suggested to have arisen in the ancient Near East. However, the current distribution of E1b1b1 in North Africa could also be caused by the back migration from the Near East to Africa that have already been proposed by several authors (Hammer et al. Genetics 1997, 145:787–805; Hammer et al. Mol Biol Evol 1998, 15:427–441; Hammer MF, et al. Mol Biol Evol 2001, 18:1189–1203). The high frequencies of haplogroup R1-M173 in Cameroon also supported the back migration from Eurasia to Africa in Cruciani et al. Am J Hum Genet 2002, 70:1197–1214. Since it's still unclear whether E1b1b evolved in Northeast Africa or the Near East, we do not attempt to interpret the presence of the two haplogroups as evidence for distinct population affinities for our three Mummy samples and we removed that argument from our study.

Q10. Line 144. “The finding of a continuous population...” Is it really a continuous population, or are the foreign influences in the Ptolemaic-Roman period probably too little to be accounted?? Would it be possible at all to trace down foreigners with this approach?? Please comment!

Answer:

The method used for the population continuity test relies on a set of priors such as generation time and total population size in the respective area. We evaluated a set of different population sizes as priors in our analysis as well as different generation times to test whether these substantially change our initial analysis. They did not change our initial finding of genetic continuity, which is why we believe that our results hold even for a larger number of parameters that could be applied here.

The test of population continuity is not capable/designed to detect outliers. In general, the level of genetic resolution (mostly mitogenomes) does not allow us to identify ‘foreigners’, mostly due to the lack of comparable population data. However, we would be able to spot outliers (i.e. potential foreigners) from our nuclear DNA data via PCA and/or Admixture. The data from our three successfully genotyped samples do suggest that they belonged to the same population.

Q11. Conclusions. What are the particular challenges of working with ancient Egyptian mummies? (see comment 3.). It seems that there was some work published on ancient pathogen DNA in Egypt, but very little on human DNA. What is the reason? Was it mainly a methodological problem, such as the non-availability of high-throughput sequencing and enrichment or are there particular issues with contamination or inhibition? I think it could be interesting for the reader to understand the potential and obstacles of working with Egyptian mummies.

Answer:

Please see answers to Q3 and Q8, and our revised introduction.

Q12. Conclusions. The authors should discuss the possible influence and presence of foreigners at the archaeological site Abusir el-Meleq and how their approach could be used to detect it. Although there appears to be a continuity in the overall population, there could be maybe some single individuals that originate from the Ptolemaic or Roman population. Would it be possible to identify those people and could this help to understand the use of certain mummification practices?

Answer:

We have substantially extended the introduction (lines 111-137) and discussion (lines 361-385, 420-432) to address to point. The data from our three successfully genotyped samples do not

suggest a foreign origin, so can be considered representing the local community profile. The contextual evidence does not allow a detailed examination of variations in mummification practices.

Q13. Conclusions. I agree that in other sites, especially in the south of Egypt, there could be a much higher influence of Sub-Saharan populations. It is well known that there were close trading connections to Nubia and other Sub-Saharan areas during the Middle and New Kingdom. What does this mean for the interpretation of your findings?

Answer:

We have addressed this point in our revised discussion (see lines 395-405). As an alternative explanation this would only mean that modern-day Egyptians might resemble more closely ancient Egyptians from the south. However, in the absence of data from southern sites, this also remains speculative.

Reviewer #2:

Q1: This paper is written like similarly structured ancient DNA papers, with a focus on the production of the dataset, a set of standard analyses performed for small sample sizes and extrapolation to prehistorical conclusions. The problem with this approach is that for the majority of studies, prior genetic analysis (and archaeological etc) has already been performed. The ancient DNA papers largely ignore the other fields and are written 'hypothesis-free.' Their scientific value becomes limited as each new paper attempts to devise and answer hypotheses that are often longstanding, involving significant repetition. This paper should and could be re-written to answer a question. It would start with a review of prior historical and genetic work performed on Egyptian populations, narrow down to 1-2 specific hypotheses (Can we confirm that slave trade into Egypt and Near East occurring in the past XX centuries changed the genetic composition of Egyptians represented by urban cohorts in Cairo?) Is this a paper about success in sequencing ancient Egyptians or about resolving the population history of Egyptians given new mtDNA data?

Answer:

As in our previous version, we had clearly framed the aims of our study: i) a proof of concept that Modern Sequencing technology produces data of previously problematic samples, ii) to evaluate the degree of continuity between the ancient and modern Egyptian population, [and iii)....]. As such, the article is not hypothesis-free. Neither are we ignorant of other fields. In

particular, in the revised manuscript we have extensively included information from historical and archaeological sources (see Introduction and Discussion).

Q2: Unfortunately, the authors appear to have ignored several earlier papers on Egyptians (see below for 3 examples, there may be others). Their primary result, stated in the abstract as “Our analyses reveal that ancient Egyptians shared more ancestry with Near Easterners than present-day Egyptians, who received additional Sub-Saharan admixture in more recent times” was shown in 2012 using genome-wide SNP array data from Egyptians (Henn et al., PLoS Genetics: Table 1). That is, the sub-Saharan African ancestry is of recent origin in Egyptians dating to AD 1250 (~24 generations ago). Indeed those authors argued that the source of this sub-Saharan ancestry is more likely to be Nilotic than West African as modeled in the current analyses. This is especially problematic as it is the only result reported in the Abstract itself. And again in the conclusion: “By comparing ancient with modern Egyptians, we remarkably found an influx of additional Sub-Saharan African ancestry after the Roman Period, which is visible at both mitochondrial and autosomal level.” This result is not remarkable, it is what is expected given prior published analyses.

Answer:

We acknowledge the oversight not having mentioned the 2012 study by Henn and colleagues, which is now included in our revised manuscript. We confirm their conclusions based on our modern Egyptian data, in particular do we confirm a time estimate for pulse-like admixture from Sub-Sahara Africa 700 years ago. However, the LD-based method used in Henn et al. 2012 (and confirmed by us) cannot distinguish between a pulse-like admixture and continuous migration. Using the direct observation from 2,000 year old Egyptians, which we estimate to have had 8% African ancestry, we get a more complete picture of the time course of African migration into the region than what can be extrapolated from modern data alone.

Q3: For example, Kujanova et al. (2009) analysis of mtDNA and Y-chromosomal data from an Egyptian oasis isolate from middle Egypt is not cited. They found that sub-Saharan mtDNA L-lineages constituted 30% of their dataset, indicating a strong sub-Saharan component on the maternal side, which was largely absent on the Y (6% M2-derived lineages). Kujanova et al. also argued for a recent migration event to account for the L lineages: “The absence of any signs of local accumulation of diversity in the L haplotypes in el-Hayez seems to favor the idea of these lineages being recent introductions into the Egyptian Western Desert.”

Answer:

We thank reviewer 2 for pointing out this dataset, which we’ve now included in the analysis and interpretation of our revised manuscript. We merged the dataset with the previously used

one by Pagani et al. (2015, AJHG). Nevertheless, the resulting statistics did only show effects at the 3rd decimal point and hence did not alter or affect our interpretation. We see this as further evidence that the origin of the dataset does not alter the statistics and the available datasets cover modern Egyptian diversity quite well.

Q4: And in Pagani et al. (2015, AJHG) show that "the average proportion of non-African ancestry in the Egyptians to be 80% and dated the midpoint of the admixture event by using ALDER to around 750 years ago (Table S2), consistent with the Islamic expansion and dates reported previously."

Answer:

Our results are consistent with a model of ancient Egypt being predominantly of near Eastern ancestry, with about 10% African ancestry, which increased up to about 18% in modern times. As we now clearly state, this is qualitatively in line with the Henn et al. study from 2012. On the level of a mixture proportion of 80% non-African ancestry, we agree with Pagani et al., but we clearly show that 2,000 years ago, Egypt was *more* Near-Eastern, not *less*, as proposed by the model in Pagani et al. with non-African ancestry appearing more recently. We discuss this in the Discussion section (see line 406 to 419).

Q5: Section beginning line 144: The authors report apparent congruity between their population sizes estimates and those from the Fayum in the Ptoleimac Period with a 95% credible interval range between 50,000 - 280,000. 1) This is the Ne estimated from mtDNA. In order to compare with the true Ne it need to be scaled 4-fold. 2) The Y-axis scale on Figure 3D does not match the results reported in the text. Indeed the Ne appears to be between 1-2 million durin the Ptolemaic period (black line). 3) Assuming even that the numbers in the text are correct, (115,000 50th estimate) the scaling for females vs. female+male as reported in the historical record would indicate a large divergence between the genetic and historical estimates. 4) No prior on the mutation rate, which will strong influence Ne, is given in the supplement. This is the appropriate place to integrate assumption of g (generation time).

Answer:

The apparent discordance between Figure 3D and the numbers given in the text stems from the fact that the figure shows unscaled results of the SkyGrid inference (given in the female effective population size times the generation time; $N_e \times T$). To estimate the approximate effective population size for the studied population, which could be compared with the population size from the historical records, we scaled the estimated values by dividing them by generation time (we assumed 25 years) and doubled them to account for the whole (male and

female) population (assuming equal male to female N_e ratio). We agree that this rescaling was not described explicitly enough, which might have made the results unclear. We modified the figure caption (figures 3d, S2) as well as the relevant methods' section so that the rescaling is more clearly described.

Regarding the mutation rate prior, we used an uninformative CTMC prior and thus all the temporal information in the inference (which drives estimation of mutation rate and the codependent parameters) came solely from ages of the samples. We modified the text accordingly to make this more clear.

Q6: Line 197: Lazaridis et al. (2014) contained 18 Egyptian sample of uncertain provenance (Cairo-urban area). Other autosomal SNP array datasets at least including many more individuals are not used here. Contrasting 3 individuals to 18 contemporary ones in order to indicate extrapolate evidence for a pattern that "6 and 7 million Sub-Saharan slaves to Northern Africa over a span of some 1250 years" seems a strong extrapolation on its own. As the authors admit.

Answer:

We cannot easily include SNP data sets that were generated on a different platform than the Affymetrix Human Origins array. But given the fact that we almost exactly confirm results from Henn et al. who used a different data set, we are confident that as far as our modern Egyptian data is concerned, we have a good representation. As far as ancient Egypt is concerned, we indeed have only three samples from a specific region, but we acknowledge the limits of our conclusions clearly in the Discussion. We also highlight that we do incorporate 100 modern Egyptian Mitogenomes from Pagani et al. and exactly confirm their report of 80% non-African ancestry at the mitochondrial level, again suggesting that our modern Egyptian sample is broadly representative at the level of the analysis performed here.

Q7: Line 180: "This individual had a derived allele at the SLC24A5 locus, which encodes for light skin pigmentation..." SLC24A5 is among at least 20 skin pigmentation genes known to contribute to melanin variability in contemporary human populations. It alone does not 'encode' for light skin pigmentation. Indeed, many individuals in Cape Verde, Africa carry this allele (which does lighten pigmentation, accounting for 7% of the variance in pigmentation) but their overall phenotype would be still be twice as dark as an average European (Beleza et al. 2013, PLoS Genetics).

Answer:

We changed the sentence (lines: 346-348) to “This individual had a derived allele at the SLC24A5 locus, which **contributes** to **lighter** skin pigmentation and was shown to be at high frequency in Neolithic Anatolia (40), consistent with the ancestral affinity shown above”

Q8: Note 8: Test of Population Continuity: the analysis here was not described. Other than collapsing mtDNA lineages into haplogroup frequencies to compare ancient and contemporary groups, there is no description of what the actual test was. Even if the method was described in Brandt et al. (2013) [not even in the main text, only in their supplement], the authors should lay out the assumptions, parameter choices and models invoked in using this method. Why for example, is TPC preferable over Approximate Bayesian Computation models typically used to test the relative likelihood of two different population demographics (in this case continuity w/ minimal drift vs. migration).

Answer:

We have extended the description of our analysis both in the methods part of the manuscript and our supplementary information for clarification and to explain our main findings. Our intention to use the TPC as applied in Brandt et al. 2013 was to evaluate with a simple method whether we can assume genetic continuity (null hypothesis) between our ancient groups and modern-day populations. We agree that complex ABC models would have been the ideal choice to explore alternative scenarios that could explain discontinuity under varying parameters (drift, migration, time, etc.), but were not deemed necessary given that we can more reliably estimate the origin and timing of admixture with nuclear data.

Reviewer #3:

Q1: I’m less confident that this result is surprising given the archaeological and historical knowledge of the region. I’m not a specialist of that history, but I would have enjoyed reading this paper much more if the introduction and discussion / conclusion would relate this finding better with the existing literature on the Egyptian history. Just stating that “Egypt provides a privileged setting for studying population history” fails a bit short of discussing open questions the study of Egyptian mummies might help to settle. Indeed, the way the paper is pitched now the real emphasis is just put on the fact that DNA could be obtained from mummies.

Answer:

We have substantially rewritten the introduction, sample collection and discussion to include more background information on Egypt’s history (lines: 57-69), the archaeological site (lines: 111-137, 361-385) and the sample collection (lines: 149-192) in particular.

Another aspect that I feel is missing is a brief discussion on how the bias from only studying individuals that were mummified (as compared to random individuals from the ancient Egypt) might have led to the lack of a Sub-Saharan component in the ancient individuals.

Answer:

We agree that taphonomic and collection bias could indeed be a factor influencing the interpretation of our results. Please see our revised discussion for reasons why we think that our sample is largely representative of the local community at the time (lines: 372-385). Of note, mummification was already quite common from the late Period on and we thus expect mainly middle class individuals at this site.

I just found one type: L123: three (instead of tree)

Answer:

Thanks, we have fixed it.